# Bony Fish Arachidonic Acid 15-Lipoxygenases Exhibit Different Catalytic Properties than Their Mammalian Orthologs, Suggesting Functional Enzyme Evolution during Vertebrate Development

**DOI:** 10.3390/ijms241814154

**Published:** 2023-09-15

**Authors:** Sophie Roigas, Kumar R. Kakularam, Michael Rothe, Dagmar Heydeck, Polamarasetty Aparoy, Hartmut Kuhn

**Affiliations:** 1Department of Biochemistry, Charité-Universitätsmedizin Berlin, Corporate Member of Freie Universität Berlin and Humboldt Universität zu Berlin, Charitéplatz 1, 10117 Berlin, Germany; sophie.roigas@charite.de (S.R.); kumar.1416@gmail.com (K.R.K.); dagmar.heydeck@charite.de (D.H.); 2Lipidomix GmbH, Robert-Rössle-Straße 10, 13125 Berlin, Germany; michael.rothe@lipidomix.de; 3Department of Humanities and Sciences, Indian Institute of Petroleum and Energy, Visakhapatnam 530003, India; aparoy@gmail.com

**Keywords:** eicosanoids, lipoxygenase, enzyme evolution, vertebrates, mammals

## Abstract

The human genome involves six functional arachidonic acid lipoxygenase (*ALOX*) genes and the corresponding enzymes (ALOX15, ALOX15B, ALOX12, ALOX12B, ALOXE3, ALOX5) have been implicated in cell differentiation and in the pathogenesis of inflammatory, hyperproliferative, metabolic, and neurological disorders. In other vertebrates, ALOX-isoforms have also been identified, but they occur less frequently. Since bony fish represent the most abundant subclass of vertebrates, we recently expressed and characterized putative ALOX15 orthologs of three different bony fish species (*Nothobranchius furzeri*, *Pundamilia nyererei*, *Scleropages formosus*). To explore whether these enzymes represent functional equivalents of mammalian ALOX15 orthologs, we here compared a number of structural and functional characteristics of these ALOX-isoforms with those of mammalian enzymes. We found that in contrast to mammalian ALOX15 orthologs, which exhibit a broad substrate specificity, a membrane oxygenase activity, and a special type of dual reaction specificity, the putative bony fish ALOX15 orthologs strongly prefer C_20_ fatty acids, lack any membrane oxygenase activity and exhibit a different type of dual reaction specificity with arachidonic acid. Moreover, mutagenesis studies indicated that the Triad Concept, which explains the reaction specificity of all mammalian ALOX15 orthologs, is not applicable for the putative bony fish enzymes. The observed functional differences between putative bony fish ALOX15 orthologs and corresponding mammalian enzymes suggest a targeted optimization of the catalytic properties of ALOX15 orthologs during vertebrate development.

## 1. Introduction

Lipoxygenases (ALOX-isoforms) are fatty acid dioxygenases catalyzing the introduction of molecular dioxygen into the hydrocarbon backbone of polyunsaturated fatty acids [1,2,3]. They carry a transition metal ion at their active site and the catalytic cycle involves redox shuttling of the valency state of this ion [1,2,3,4]. ALOX-isoforms are widely distributed in higher plants [1,3] and animals [2,5] including bony fish [6,7,8], but they occur less frequently in lower organisms [9]. In bacteria, these enzymes rarely occur [10,11] and no functional ALOX-isoforms have been described in *Archaea* [11].

In mammals, ALOX-isoforms are widely distributed and several ALOX-isoforms are simultaneously expressed in most mammalian species. The human genome involves six functional *ALOX* genes (*ALOX15*, *ALOX15B*, *ALOX12*, *ALOX12B*, *ALOX5*, *ALOXE3*) and each of them encodes for a distinct ALOX-isoform [12,13]. In the mouse reference genome, a single orthologous gene (*Alox15*, *Alox15B*, *Alox12*, *Alox12B*, *Alox5*, *Aloxe3*) exists for each human ALOX gene, but in addition a functional *Aloxe12* gene has been identified [13]. Knockout studies of the different mouse *Alox* genes [14,15,16,17,18,19] indicated that the genomic multiplicity may not be considered an indicator of functional redundancy. In fact, each mouse Alox-isoform exhibits its own functional profile.

In contrast to our comprehensive knowledge on mammalian ALOX-isoforms, little is known about the ALOX pathway in non-mammalian vertebrates. Bony fish represent the most abundant class of extant non-mammalian vertebrates and the occurrence of ALOX-isoforms in these animals has previously been reported. For instance, in 1986 [20], an ALOX activity was reported in trout gill tissue and this enzyme oxygenated arachidonic acid to its 12-hydro(pero)xy derivative (12-H(p)ETE). Later on, specific ALOX products were detected in different organs of rainbow trout [21]. ALOX-isoforms were also detected in macrophages [6] and blood platelets [7] of other trout species. Peripheral leukocytes of the Atlantic salmon and the mirror carp produced specific ALOX products when stimulated in vitro with calcium ionophore [22] and similar results were obtained for peripheral blood cells of the lesser spotted dogfish [23]. In 2016, an ALOX5 ortholog of the large yellow croaker (*Larimichthys crocea*) was cloned [24], but it remained unclear how frequently ALOX15 orthologs do actually occur in bony fish.

The zebrafish constitutes a model organism for bony fish, which is frequently employed to explore mechanistic details of vertebrate embryogenesis [25,26]. In 2013, the reference genome of the zebrafish was published [27] and a detailed search of the genomic sequence database indicated the presence of seven ALOX genes (*zbfALOX1*, *zbfALOX2*, *zbfALOX3a-c*, *zbfALOX4a-b*). Unfortunately, detailed information on the enzymatic properties of the corresponding ALOX-isoforms are only available for zbfALOX1 [28] and zbfALOX2 [29]. Zebrafish ALOX2 was characterized as arachidonic acid (AA) 5-lipoxygenating enzyme capable of producing leukotrienes [29] and comparison of the gene structures of the zebrafish LOX2 gene with the *ALOX5* genes of humans and mice suggested that zebrafish LOX2 is the zebrafish ortholog of human and mouse ALOX5. zbfALOX1 was characterized as AA 12-lipoxygenating enzyme and preliminary functional characterization of the recombinant enzyme suggested pronounced catalytic differences to mammalian ALOX15 orthologs [28]. Expression silencing of this enzyme compromised cerebral embryogenesis [30] and this data was inconsistent with the previous observation that *Alox15^−^*^/*−*^ mice develop normally [15].

We recently expressed putative ALOX15 orthologs of three different bony fish species and characterized basic catalytic properties [8]. To explore whether these enzymes constitute functional equivalents of mammalian ALOX15 orthologs, we here determined additional functional properties (enantioselectivity, substrate specificity, membrane oxygenase activity) of the bony fish enzymes and compared selected structural and functional enzyme characteristics of these proteins with those of the corresponding mammalian proteins. Despite the high degree of structural similarity between the putative bony fish ALOX15 orthologs and the mammalian enzymes, we observed pronounced functional differences. Moreover, results of mutagenesis studies suggested that the Triad Concept [31] explaining the reaction specificity of all mammalian ALOX15 orthologs tested so far is not applicable to the putative bony fish enzymes. Taken together, our data suggest a functional optimization of ALOX15 orthologs during vertebrate evolution.

## 2. Results

### 2.1. Database Searches Suggested the Presence of ALOX15 Genes in Bony Fish

When we searched the NCBI protein database for annotated ALOX15 orthologs and ran the positive hits through our ALOX filtering strategy (see Section 4), we detected corresponding sequences in seven different bony fish species (Table 1).

Since the sequence for *Takifugu rubripes* was incomplete and the sequence for *Hippocampus comes* lacked the iron ligand clusters (Appendix A), they were disclosed from further investigations. The remaining hits involved the zebrafish LOX1, which was previously expressed as recombinant protein and characterized as arachidonic acid (AA) 12*S*-lipoxygenating enzyme [28,30].

To quantify the degree of structural similarity of the remaining five putative bony fish ALOX15 orthologs, we carried out amino acid alignments and these data indicated that the bony fish enzymes share a high (71–93%) degree of amino acid conservation (Appendix A). Then we compared the amino acid sequences of the putative bony fish ALOX15 orthologs with those of human (Appendix A) and mouse ALOX15 (Appendix A). Here we found that the degrees of amino acid sequence identity were much lower [ranging between 41 and 48% (Appendix A)]. 

Finally, we compared the structures of the putative bony fish ALOX15 genes with those of the human ALOX-isoforms (Figure 1) and found that with the exception of ALOX5 the size of human ALOX genes varied between 10 and 23 kbp. The genes of the putative bony fish ALOX15 orthologs have similar lengths (Figure 1). On the other hand, the human *ALOX5* gene was much bigger. All putative bony fish ALOX15 genes consist of 14 exons and 13 introns and most human ALOX genes (ALOX15, ALOX15B, ALOX12, ALOX5) share a similar exon-intron-composition. However, human ALOXE3 and ALOX12B involve 15 exons and 14 introns and these data suggested a lower degree of structural similarity of the putative bony fish ALOX15 orthologs with these human ALOX genes. The degree of nucleotide sequence identity of the putative bony fish ALOX15 genes with that of the human ALOX-isoforms varied between 22 and 40% (Table 2). 

The putative bony fish ALOX15 genes share the highest degree of nucleotide identity with human ALOX12 (38%), but the degree of sequence identity with human ALOX12B and ALOXE3 was not much lower. Taken together, on the basis of these sequence data, it was not possible to precisely assign the putative bony fish ALOX15 orthologs to any of the human isoforms. Since except for the human ALOX5 gene, all other human ALOX genes are localized in a joint gene cluster on the short arm of chromosome 17 [12], chromosomal localization was neither helpful for such assignment.

Next, we compared the 3D structures of the putative bony fish ALOX15 orthologs with human ALOX15B, rabbit ALOX15, and human ALOX12. For this purpose, we first generated homology 3D models for the three bony fish enzymes on the basis of the X-ray coordinates of human ALOX15B (PDB: 4NRE) and rabbit ALOX15 (PDB: 2P0M) and tested the validity of these models using two different software tools (see Section 4.3.) Then we overlaid the bony fish models with the crystal structure of human ALOX15B [32], rabbit ALOX15 [33], and a home-made model of human ALOX12.

From Figure 2, it can be seen that the putative ALOX15 of *N. furzeri* adopts a similar 3D structure as human ALOX15B, rabbit ALOX15, and human ALOX12. The canonical ALOX-fold (two domain structure) is visible and the spacial relations of the secondary structural elements are almost identical. The high degree of 3D structural similarity of the bony fish ALOX-isoforms with the human enzymes is also indicated by the calculated similarity scores.

When the 3D models of the putative bony fish ALOX15 orthologs were constructed on the basis of the X-ray coordinates of human ALOX15B (Table 3, upper part), the overall structural similarity scores of the bony fish models with rabbit ALOX15, human ALOX15B, and human ALOX12 varied between 1 and 3, and this data indicates that all enzymes share a high degree of structural similarity. The lower values (higher degree of structural similarity) obtained for human ALOX15B are due to the fact that, for this comparison, the bony fish models were constructed on the basis of the X-ray coordinates of this enzyme (PDB: 4NRE). When similar structural comparisons were carried out with bony fish 3D models constructed on the basis of the X-ray coordinates of rabbit ALOX15 (2P0M), lower similarity scores were obtained for this enzyme (Table 3, lower part). Here again, the similarity scores for all bony fish enzymes varied between 1 and 3. For better interpretation of the structural similarity scores, we also compared the degree of structural similarity among different mammalian ALOX-isoforms. Here, the numeric values varied between 2 and 4 (humALOX15B vs. rabALOX15: 3.155, hALOX15B vs. hALOX5: 2.315, hALOX5 vs. rabALOX15: 3.213). On the other hand, when similar calculations were carried out using the X-ray coordinates of *Pseudomonas aeruginosa* ALOX, much higher values (lower degree of structural similarity) were obtained: humALOX15B: 16.256, rabALOX15: 9.114, humALOX5: 8.409). 

Taken together, on the basis of these structural modeling data, it was not possible to assign the putative bony fish ALOX15 orthologs to any of the human ALOX-isoforms. In other words, on the basis of all structural data we collected, it was not possible to unequivocally assign the putative bony fish ALOX15 orthologs to any of the mammalian ALOX-subfamilies and thus, we decided to functionally characterize the bony fish enzymes in more detail.

### 2.2. The Putative Bony Fish ALOX15 Orthologs Are Well Expressed in Insect Cells

For functional characterization, we first expressed the putative bony fish ALOX15 orthologs as N-terminal his-tag fusion proteins in Sf9 insect cells. From Figure 3, it can be seen that the three putative ALOX15 orthologs are well expressed in insect cells.

When we quantified the immune signals densitometrically and employed the signal intensity of the pure *Myxococcus fulvus* ALOX as calibration standard, we estimated the expression levels of the three putative bony fish ALOX15 orthologs. As indicated in Table 4, the putative ALOX15 ortholog of *N. furzeri* was expressed at a level of about 5 mg recombinant protein per liter bacterial liquid culture. For the other recombinant proteins, somewhat lower expression levels were achieved, but they were also sufficient for functional enzyme characterization.

### 2.3. The Putative Bony Fish ALOX15 Orthologs Convert Arachidonic Acid to 12S- and 8R-HETE

In previous experiments [8], we observed that the putative bony fish ALOX15 orthologs convert arachidonic acid with dual reaction specificity to variable amounts of 12- and 8-HETE, but the enantiomer composition of the major reaction products has not been explored. To fill this gap, we carried out in vitro activity assays using arachidonic acid as substrate and first analyzed the conjugated dienes formed by RP-HPLC. 

From Figure 4, it can be seen that the three enzymes convert AA dominantly to conjugated dienes, which co-chromatograph in RP-HPLC with authentic standards of 12-HETE and 8-HETE (Figure 4B,D,F). Both compounds were not well resolved under our chromatographic conditions. When the incubations were carried out in the absence of enzyme (Figure 3A), with heat inactivated cell lysis supernatants (Figure 4C) or with a lysis supernatant prepared from Sf9 cells infected with a non-lipoxygenase control baculovirus (Figure 4E), formation of these products was not detected. 

Next, the conjugated dienes formed (Figure 5) were prepared by RP-HPLC and further analyzed by combined normal phase/chiral phase HPLC (NP/CP-HPLC). When we analyzed the oxygenation products formed by the putative ALOX15 ortholog of *N. furzeri*, the peak eluted with a retention time of about 10 min in RP-HPLC was split into two separate peaks co-eluting with standards of 12*S*-HETE and 8*R*-HETE (Figure 5A). For this enzyme, 12*S*-HETE was clearly dominant, since more than 80% of the conjugated dienes formed were identified as 12*S*-HETE. 12*R*-HETE and 8*S*-HETE were virtually absent and this data indicates that oxygen insertion into the fatty acid substrate was fully enzyme controlled. When similar analyses were carried out with the oxygenation products formed by the putative ALOX15 of *P. nyererei*, the same oxygenation products (12*S*-HETE and 8*R*-HETE) were detected, but here, 8*R*-HETE contributed almost one third to the sum of the AA oxygenation products (Figure 5B). For the enzyme of *S. formosus*, the relative share of the major oxygenation product (12*S*-HETE) was down to about 60% (Figure 5C). Other HETE isomers, such as 15*R*/*S*-HETE, 11*R*/*S*-HETE, and 5*R*/*S*-HETE, which are also well separated in this chromatographic system, were not detected. For independent evidence for the chemical structure of the major oxygenation product formed by the putative bony fish ALOX15 orthologs, we performed LC-MS/MS analysis of the 12S-HETE peak prepared by NP/CP-HPLC. In our LC-MS/MS system, the 12S-HETE peak co-eluted with an authentic standard of 12S-HETE and the major fragmentation ions at *m*/*z* 319 (M^+^), *m*/*z* 301 (M^+^-water), *m*/*z* 257 (M^+^-water—carbon dioxide) are consistent with the supposed chemical structure (Appendix A). The major alpha-cleavage ion at *m*/*z* 179 indicates the position of the OH-group at C12. The loss of carbon dioxide from the *m*/*z* 179 alpha cleavage ion leads to the formation of the *m*/*z* 135 ion (Appendix A).

### 2.4. The Putative Bony Fish ALOX15 Orthologs Incorporate Atmospheric Oxygen during Both 12S-HETE and 8R-HETE Formation

When oxidizing polyenoic fatty acids, ALOX-isoforms incoporate one molecule of atmospheric oxygen into the primary oxygenation products. To confirm this reaction mechanism for the putative bony fish ALOX15 orthologs, we carried out activity assays under ^18^O_2_ atmosphere and determined the percentage of ^18^O-incorporation into the reaction products.

For this purpose, the reaction products were prepared by combined NP/CP-HPLC and the ratios of the mass ions at *m*/*z* 319 (incorporation of one ^16^O atom in the reaction product) and *m*/*z* 321 (incorporation of one ^18^O atom into the reaction product) were quantified for both, 12*S*- and 8*R*-HETE by LC-MS/MS (Appendix A). The data presented in Table 5 indicate that the two major reaction products are formed via the incorporation of atmospheric oxygen. The small share of ^16^O-incorporation may be related to incomplete removal of ^16^O_2_ from the reaction mixture and to the fact that the enzyme solutions could not completely be freed from atmospheric oxygen. Our attempts to anaerobize the enzyme solutions led to complete loss of the catalytic activity. Thus, the experiments were carried out with aerobic enzyme solutions, which introduces ^16^O_2_ into the incubation mixture.

### 2.5. The Putative Bony Fish ALOX15 Orthologs Strongly Prefer C_20_ Fatty Acids

Mammalian ALOX15 orthologs accept most naturally occuring polyenoic fatty acids, but this is not the case for other ALOX-isoforms. For instance, ALOX12 orthologs from different mammals strongly prefer C_20_ fatty acids [34]. These enzymes also oxygenate C_22_ fatty acids, but do not accept polyunsaturated fatty acids (PUFAs) lacking an n-11 bisallylic methylene, such as linoleic acid (LA) or alpha-linolenic acid (α-LnA). However, they accept gamma-linolenic acid (g-LnA) as substrate, which carries an n-11 bisallylic methylene [34]. Similarly, human ALOX15B prefers AA as substrate over linoleic acid [35]. To test the substrate specificity of the putative bony fish ALOX15 orthologs, we compared the capability of these enyzmes to oxygenate a number of naturally occurring PUFAs.

From Figure 6, it can be seen that the C_20_ fatty acids arachidonic acid (AA) and eicosapentaenoic acid (EPA) are strongly prefered as substrates when compared to docosahexaenoic acid (DHA), linoleic acid (LA), alpha-linolenic acid (α-LnA), or gamma linolenic acid (γ-LnA), and these findings suggest a functional difference to mammalian ALOX15 orthologs. In an additional set of experiments, we offered an equimolar mixture of AA, EPA, and DHA (17 µM each) as substrate for the putative bony fish ALOX15 orthologs. Here again, we observed that the C_20_ fatty acids (AA and EPA) were preferentailly oxygenated and that DHA was only a weak substrate (Appendix A).

### 2.6. The Putative Bony Fish ALOX15 Orthologs Lack Biomembrane Oxygenase Activities

Mammalian ALOX15 orthologs are capable of oxygenating complex PUFA-containing ester lipids, even if these substrates are present in biomembranes [36,37]. To test whether the putative bony fish ALOX15 orthologs are also capable of oxygenating biomembranes, aliquots of the enzyme preparations were incubated with submitochondrial membranes.

For comparative purposes, control incubations were carried out with recombinant rabbit ALOX15. To strictly compare the membrane oxygenase activities, we first normalized the AA oxygenase activities of the different enzyme preparations. Next, identical AA oxygenase activities were incubated in PBS with submitochondrial membranes (1.4 mg membrane protein/mL) for 15 min. The reaction products were reduced, the total lipids were extracted, hydrolyzed under alkaline conditions, and the hydrolysates were analyzed by RP-HPLC. Recording the chromatograms at 235 nm, the oxygenated PUFAs (HODE and HETE isomers) were quantified. At 210 nm, we analyzed the non-oxygenated PUFAs. Applying this experimental strategy, we were able to quantify the OH-PUFA/PUFA ratio before and after the incubation period, which is a suitable measure for the oxidation degree of the membrane lipids.

In Figure 7, a representative RP-HPLC chromatogram of the hydrolyzed membrane lipid extracts of rabbit ALOX15 treated submitochondrial membranes is shown. Following the chromatogram at 235 nm (Figure 7A), we observed the presence of conjugated dienes (inset), which co-chromatographed with authentic standards of 13-HODE and 15-HETE. These products were absent in non-enzyme control incubations and thus, must have been formed during ALOX15 membrane interaction. When we recorded the chromatogram at 210 nm (Figure 7B), we observed two major peaks, which co-eluted with authentic standards of arachidonic acid (AA) and linoleic acid (LA). From these chromatographic data, we calculated a hydroxy-PUFA/PUFA ratio of about 5% (Table 6), indicating that one out of 20 PUFA molecules were present in the membrane phospholipids as oxygenated derivatives. When similar incubations were carried out with the enzyme preparations of the putative bony fish ALOX15 orthologs, we obtained much lower OH-PUFA/PUFA ratios (Table 6) and these data suggest that the biomembrane oxygenase activity of the bony fish enzymes is limited. 

### 2.7. The Triad Concept Is Not Applicable to the Putative Bony Fish ALOX15 Orthologs

The Triad Concept was developed to explain the structural basis for the dual reaction specificity of mammalian ALOX15 orthologs and recent experiments indicated that this concept is applicable to all mammalian ALOX15 orthologs tested so far [31]. This concept suggests that the site chain geometry of the triad determinants (Phe353, Ile418 + Met419, Ile593 of human ALOX15) is important for the reaction specificity of these enzymes. If amino acids with bulky side chains are localized at these positions, AA 15-lipoxygenation is catalyzed. In contrast, if these amino acids carry small side chains, 12-HETE is dominantly formed [31].

When we aligned the amino acid sequences of the putative bony fish ALOX15 orthologs and compared these sequences with that of human and rabbit ALOX15, we found that Phe353 of human ALOX15 is conserved in the putative bony fish ALOX15 orthologs. In contrast, the Ile418 + Met419 motif of human ALOX15 is replaced by a Val + Val motif (Appendix A), which is cannonic for AA 12-lipoxygenating ALOX15 orthologs. Thus, formally the putative bony fish ALOX15 orthologs follow the Triad Concept. However, the Triad Concept also suggests that site-directed mutagenesis of the triad determinants must alter the reaction specificity of the enzymes [31,39]. To test this hypothesis, we carried out corresponding mutagenesis experiments and mutated the Val422/424 + Val423/425 motif in the three putative bony fish ALOX15 orthologs to Ile422/424 + Met423/425 and quantified the reaction specficity of the double mutants. If the Triad Concept is applicable for the putative bony fish ALOX15 orthologs, 15-HETE was expected as a major AA oxygenation product.

However, except for the observation that the double mutants exhibit a reduced catalytic activity, the product pattern of the double mutant (Figure 8B) was similar to that of the wildtype enzyme (Figure 8A). Most importantly, we did not observe a major increase in the formation of 15-HETE. Consequently, Val425Ile + Val425Met exchange hardly altered the reaction specificity of the enzyme. Similar chromatograms were also obtained for the Val + Val to Met + Ile double mutants of the putative ALOX15 orthologs of *P. nyererei* and *S. formosus* (Appendix A). For all mutants, the 8/12-HETE peak was dominant. These results indicate that the Triad Concept is not applicable for the bony fish enzymes and thus, these proteins are functionally distinct from their mammalian counterparts.

### 2.8. The Putative Bony Fish ALOX15 Orthologs Partly Follow the Gly-vs.-Ala Concept

The carbon atom of the fatty acid that is oxygenated during the ALOX reaction is converted to a chiral center and thus, in principle, two different enantiomers (e.g., 12S-HETE vs. 12R-HETE) can be formed. Multiple sequence alignments of several ALOX-isoforms suggested that *S*-lipoxygenating enzymes carry an Ala at a critical position (Coffa determinant) of their primary structure [40,41]. In contrast, in *R*-lipoxygenating enzymes, a Gly residue is localized at this position [40,41]. Ala-to-Gly exchange converts S-lipoxygenating ALOX-isoforms into *R*-lipoxygenating enzymes and vice versa [40,41,42]. However, LOX1 of *D. rerio* carries a Gly at this critical position (Gly410), but converts AA almost exclusively to 12*S*-HETE and thus, *D. rerio* LOX1 violates at least in part the Gly-vs.-Ala concept [28].

When we aligned the amino acid sequences of the putative bony fish ALOX15 orthologs with that of the *D. rerio* LOX-1, we found (Appendix A) that the Coffa determinant was occupied by a Gly, which immediately trails an Arg (RG motif). To explore whether amino acid exchanges in this region of the primary structure alter the reaction specificity of the putative bony fish ALOX15 orthologs, we first carried out a Gly410Ala exchange in the *N. furzeri* enzyme and expected a decrease in the relative share of the 8*R*-HETE formation. In fact, the share of 8*R*-HETE formation was reduced from 15% for the wildtype enzyme (Table 7) to 7% for the Gly410Ala mutant (Figure 9B, Table 7). Interestingly, the Arg409Ile exchange induced an opposite alteration in the reaction specificity. Here, the relative share of 8*R*-HETE increased to 40% of the sum of the HETE isomers (Figure 9C, Table 7). The putative ALOX15 of *P. nyererei* converted AA to a 1:4 mixture of 8*R*-HETE and 12*S*-HETE (Figure 5B) and this result was confirmed in our mutagenesis studies (Table 6). Gly410Ala exchange strongly augmented the relative share of 12*S*-HETE. In fact, 12*S*-HETE was the only AA oxygenation product detected (Table 7). Arg409Ile exchange did induce an increase in 8*R*-HETE formation, but here the relative share of this product was doubled. Similar alterations were observed for the putative ALOX15 from *S. formosus*. Here again, 12*S*-HETE was the only AA oxygenation product formed by the Gly410Ala mutant and an almost 1:1 mixture of 8*R*- and 12*S*-HETE was formed by the Arg409Ile mutant.

Taken together, these mutagenesis data indicate that although the Gly-vs.-Ala hypothesis does not correctly predict the reaction specificity of the wildtype putative bony fish ALOX15 orthologs, mutagenesis of the Coffa determinant and of the adjacent Arg altered the reaction specificity of the enzymes and thus, the enzymes partly follow the A-vs.-G hypothesis.

## 3. Discussion 

### 3.1. The Putative Bony Fish ALOX15 Orthologs Exhibit Different Catalytic Properties than Corresponding Mammalian Enzymes

Lipoxygenases are polyenoic fatty acid oxygenating enzymes that frequently occur in highly developed plants and animals [1,2,3]. They also occur in lower organisms, but here they are not widely distributed. In the genomes of most mammals, several ALOX genes have been detected. The human genome involves six functional ALOX genes [12,13], and one of them encodes for ALOX15. The first mammalian ALOX15 ortholog was described almost 50 years ago in immature red blood cells [36]. A recent search for ALOX15 orthologs revealed that these enzymes frequently occur in *Metatheria* and *Eutheria*, but that they have not been identified in *Prototheria* [31]. When we searched the public bony fish databases, which involve some 50 entries at the time of searching, using the key words “lipoxygenase” and “bony fish”, we obtained numerous hits, but only seven of them were annotated as ALOX15 orthologs. These data indicated that ALOX sequences frequently occur in bony fish, but that ALOX15 orthologs are rare. Multiple sequence alignments of the identified *ALOX15* genes indicated that the putative ALOX15 orthologs shared a similar degree of amino acid and nucleotide sequence conservation with all human and mouse ALOX-isoforms (Table 3, Appendix A). Moreover, 3D models of the putative bony fish ALOX15 orthologs (Figure 2), which were constructed on the basis of human ALOX15B and rabbit ALOX15, did not allow precise structural assignment of the putative bony fish ALOX15 orthologs to any of the mammalian ALOX-subclasses. The degree of structural similarity to rabbit and human ALOX-Isoforms was very similar (Table 3). Thus, based on these structural data, it was not possible to decide whether the putative bony fish ALOX15 orthologs fulfill the same in vivo functions as mouse and human ALOX15. To obtain functional data for the bony fish enzymes, we expressed three of them as recombinant N-terminal his-tag fusion proteins in Sf9 cells, characterized selected catalytic properties, and compared these enzymatic characteristics with those of mammalian ALOX15 orthologs. Taken together, these functional data suggest that the putative bony fish ALOX15 orthologs have fundamentally different catalytic characteristics:(i)The bony fish enzymes exhibit different reaction specificites: Although mammalian ALOX15 orthologs and the bony fish enzymes exhibit dual reaction specificity (Figure 4 and Figure 5), the molecular basis for this enzyme property is different. Mammalian ALOX15 orthologs convert AA to a mixture of 12-HETE and 15-HETE [31]. The formation of these products involves hydrogen abstraction from two different bisallylic methylenes (C_10_ and C_13_, respectively) and [+2] rearrangement of the carbon-centered fatty acid radical. According to the current concept of the ALOX reaction, the formation of these products involves tail-first substrate binding at the active site [43,44,45]. The putative bony fish ALOX15 orthologs convert AA to a mixture of 12*S*-HETE and 8*R*-HETE. The formation of these products involves hydrogen abstraction from a single bisallylic methylene (C_10_), but simultaneous [+2] (12*S*-HETE formation) and [−2] (8*R*-HETE formation) radical rearrangement;(ii)The bony fish enzymes exhibit a different substrate specificity than mammalian ALOX15 orthologs: Mammalian ALOX15 orthologs accept most naturally occurring polyenoic fatty acids as substrates independent of their chain length and the degree of unsaturation. Here, we found (Figure 6) that the putative bony fish ALOX15 orthologs strongly prefer C_20_ polyenoic fatty acids (AA, EPA), but hardly oxygenate C_22_ (DHA) and C_18_ PUFAs (α-LnA, γ-LnA);(iii)The bony fish enzymes lack membrane oxygenase activities: Mammalian ALOX15 orthologs are capable of oxygenating PUFAs bound in membrane phospholipids [36,37]. In contrast, the putative bony fish ALOX15 orthologs lack a sizable membrane oxygenase activity (Table 6). In fact, for the enzymes of *N. furzeri* and *S. formosus*, no membrane oxygenase activity was detected and the corresponding activity of the *P. nyererei* enzyme amounted to less than 5% of that of rabbit ALOX15 when the membrane oxygenase activity was normalized to similar AA oxygenase activities;(iv)The bony fish enzymes do not follow the Triad Concept: The reaction specificity of mammalian ALOX15 orthologs can be predicted on the basis of their primary structures and comprehensive mutagenesis studies recently confirmed the predictive value [31]. Here, we observed that the formation of 12*S*-HETE by the bony fish enzymes is consistent with the Triad Concept, but that mutagenesis of the triad determinants did not alter the reaction specificity of these enzymes (Figure 8). Thus, in contrast to mammalian ALOX15 orthologs, the Triad Concept may not be applicable for the bony fish enzymes;(v)The Gly-vs.-Ala concept is partly applicable for the bony fish enzymes: Mammalian ALOX15 orthologs are *S*-lipoxygenating enzymes and their enantioselectivity is consistent with the Gly-vs.-Ala hypothesis [41,42]. However, mutations of the Coffa determinants [40,41,42] only induced minor alterations in the reaction specificity of several mammalian ALOX15 orthologs [46]. Here, we found that the reaction specificity of the bony fish enzymes was not correctly predicted by the Gly-vs.-Ala concept (a small Gly is located at the Coffa position, but the major reaction product was 12*S*-HETE) and that Gly-to-Ala exchange augmented the relative share of 12*S*-HETE formation (Figure 9). Thus, despite the failure of the Gly-vs.-Ala concept to correctly predict the reaction specificity of the bony fish ALOX-isoforms, the enzymes partly follow the Gly-vs.-Ala hypothesis.

Taken together, our functional data indicate that major catalytic properties of the putative bony fish ALOX15 orthologs are different from those of the corresponding mammalian enzymes. Thus, the bony fish enzymes might not fulfill the same in vivo functions as their mammalian counterparts. Comparing the catalytical properties of the putative bony fish ALOX15 orthologs with those of the mammalian enzymes, one gets the impression that the functional properties of these enzymes have been optimized during vertebrate evolution. When we explored the reaction specificity of a large number of mammalian ALOX15 orthologs, we previously reported that this enzyme property has systematically been altered during mammalian evolution [31]. In that paper, we showed that mammals ranked in evolution lower than gibbons, including the frequently used lab animals (mice and rats) express arachidonic acid 12-lipoxygenating Alox15 orthologs. In contrast, hominids (different human subspecies, chimpanzee, bonobos, gorillas, orangutans) express arachidonic acid 15-lipoxygenating enzymes. Although some mammalian species (rabbits, kangaroo rats) violate this concept, more than 95% of mammalian ALOX15 orthologs do follow it. In other words, most mammalian ALOX15 orthologs are arachidonic acid 12-lipoxygenating enzymes. The putative bony fish ALOX15 orthologs described here are also arachidonic acid 12-lipoxygenating enzymes and thus, in this respect, they are more closely related to the enzymes of non-hominid mammals than to the hominid orthologs. Moreover, the bony fish enzymes exhibit a narrow substrate specificity. They strongly prefer C_20_ fatty acids, but hardly oxygenate C_22_ and C_18_ fatty acid derivatives (Figure 6). Thus, the substrate specificity of the ALOX15 orthologs was broadened during vertebrate evolution. Since C_18_ fatty acids (linoleic acid, linolenic acid) frequently occur in membranes of eucaryotic cells, bony fish ALOX15 orthologs were likely to exhibit a limited membrane oxygenase activity. On the other hand, mammalian ALOX15 orthologs are capable of oxygenating biomembranes [36,37]. Testing the membrane oxygenase activity of the three putative bony fish ALOX15 orthologs we found that the bony fish enzymes lack any membrane oxygenase activity. Thus, ALOX15 orthologs have apparently acquired a membrane oxygenase activity during vertebrate evolution. Although the driving forces for the evolutionary change in reaction specificity remain to be explored, these functional differences (broadening of substrate specificity, acquiring a membrane oxygenase activity) might have optimized the biological role of mammalian ALOX15 orthologs during the maturational breakdown of mitochondria during late erythropoiesis [36,47].

### 3.2. Degree of Novelty and Biological Relevance of the Presented Findings

The metabolism of arachidonic acid in general and the ALOX pathway in mammals have well been characterized in recent years, but there is a shortage of information on the ALOX pathway in other vertebrates. According to recent estimates of the International Union for Conservation of Nature, some 74,000 vertebrate species are currently living on earth and almost half of them are fish (Appendix A). Compared with all invertebrates (some 1.5 million different species), vertebrates only contribute a minor share (5%) to the extant animal species on this planet, but fish clearly dominate other vertebrates [48]. Since most fish species are classified as bony fish [49], we limited our search for ALOX15 orthologs to bony fish. When we started this project, only scattered information was available on the eicosanoid metabolism of bony fish [6,7,22,23,24]. These studies focused on the identification of the ALOX products, but did not characterize the corresponding enzymes. Thus, except for the LOX1 [28,30] and LOX2 [29] of *D. rerio*, there was neither structural nor functional information available on any bony fish ALOX-isoform. We recently cloned and expressed putative ALOX15-isoforms from three different bony fish species and determined a number of kinetic properties [8], but the substrate specificity, the enantioselectivity, and the membrane oxygenase activity of these enzymes have not been characterized. Moreover, no mutagenesis experiments have been carried out to test whether these enzymes follow the currently available concepts (Triad Concept, A-vs.-G-Hypothesis) explaining the reaction specificity of lipoxygenases. These gaps of knowledge have been filled by the functional data reported in this study. 

According to our data, the putative bony fish ALOX15 orthologs exhibit fundamentally distinct functional characteristics compared to mammalian ALOX15 orthologs. Thus, it is unlikely that these enzymes constitute functional equivalents of mammalian ALOX15 orthologs in bony fish. This conclusion is supported by the differential outcome of knockout studies in zebrafish. Expression knockdown of the ALOX1 gene induced defective embryo development [30]. In contrast, functional inactivation of the Alox15 gene in mice did not induce major phenotypic alterations [15]. These data suggest that the zebrafish LOX1 and the bony fish ALOX-isoforms characterized in this study, which share a high degree of evolutionary relatedness with each other, might fulfill different biological functions. Mammalian ALOX15 orthologs have been implicated in the maturational breakdown of mitochondria during late red blood cell maturation [47]. Since the putative bony fish ALOX15 orthologs lack a sizable membrane oxygenase activity (Table 5), these enzymes cannot fulfill this function. Moreover, mammalian ALOX15 orthologs have been implicated in the biosynthesis of pro-resolving lipoxins [50]. However, arachidonic acid 12- and arachidonic acid 8-lipoxygenation counteracts lipoxin biosynthesis, so the putative bony fish ALOX15 orthologs are likely to exhibit reduced capacities for the biosynthesis of pro-resolving lipoxins, although these compounds have been detected in bony fish [22].

### 3.3. Limitations of the Study

We characterized the functional properties of the putative bony fish ALOX15 orthologs using crude enzyme preparations. This limitation was related to our inability to purify the recombinant enzymes by affinity chromatography on a Ni-NTA-agarose. Although we attempted to optimize the chromatographic conditions and we employed alternative protein purification strategies (gel filtration, ion exchange chromatography), we always observed severe losses in the catalytic activity of the three enzymes. The molecular basis for this catalytic inactivation has not been explored in detail, but it may be related to storage and handling lability of the enzymes. Moreover, we observed that repeated freezing harmed the catalytic activity. Fortunately, previous reports on the catalytic activity of other ALOX-isoforms indicated that the degree of purity of enzyme preparations does hardly impact important functional properties of lipoxygenases, such as reaction specificity, substrate specificity, and membrane oxygenase activity [34,36]. Thus, experiments with crude enzyme preparations are not misleading.

When we initiated our study, genomic sequences of some 55 different bony fish species had been deposited in the database. Considering the recent estimate of the IUCN that some 36,000 fish species are currently living on earth (Appendix A), the size of the employed database is rather small and certainly not representative. However, the observation that putative functional ALOX15 genes were only present in six (Table 1) of the explored bony fish genomes suggests that such enzymes do not frequently occur in bony fish. In mammals, ALOX15 orthologs have been detected in most metatherian and eutherian species, although in some of them (guinea pig, naked mole rat), such enzymes are also lacking [31].

A major driving force of this study was the lack of functional information on bony fish ALOX15 orthologs. When we compared the amino acid sequences of the putative bony fish enzymes with the corresponding sequences of human and mouse ALOX-isoforms (Appendix A), we only observed a relatively low (40–50%) degree of amino acid conservation. Most importantly, the degree of amino acid conservation was similar when the bony fish enzymes were run against the different mouse and human isoforms. Thus, on the basis of the amino acid sequences, it was impossible to assign the putative bony fish ALOX15 orthologs to any of the human enzymes. Such assignment also failed when the genomic sequences (Table 2), the gene structures (Figure 1), and the 3D-strutures (Figure 2, Table 3) of the enzymes were employed as readout parameters. Unfortunately, our hope to unequivocally assign the putative bony fish ALOX15 orthologs to any of the human isoforms (ALOX15, ALOX15B, ALOX12, ALOX12B, ALOX5, ALOXE3) on the basis of the collected functional data did not work out. In other words, even with the help of the functional data presented here, it was impossible to assign the putative bony fish ALOX15 orthologs to any of the mouse and human ALOX-isoform classes.

## 4. Materials and Methods

### 4.1. Chemicals and Devices

The chemicals used for the different experiments were obtained from the following sources: Phosphate buffered saline without calcium and magnesium (PBS) from PAN Biotech (Aidenbach, Germany); nitrocellulose blotting membrane from Serva Electrophoresis GmbH (Heidelberg, Germany); EDTA (Merck KG, Darmstadt, Germany), arachidonic acid (AA) and authentic HPLC standards of HETE-isomers (15S-HETE, 15S/R-HETE, 12S/R-HETE, 12S-HETE, 8S/R-HETE, 5S-HETE) from Cayman Chem (distributed by Biomol GmbH, Hamburg, Germany); acetic acid from Carl Roth GmbH (Karlsruhe, Germany); sodium borohydride from Life Technologies, Inc. (Eggenstein, Germany); isopropyl-β-thiogalactopyranoside (IPTG) from Carl Roth GmbH (Karlsruhe, Germany); restriction enzymes from ThermoFisher (Schwerte, Germany); the *E. coli* strain Rosetta2 DE3 pLysS from Novagen (Merck-Millipore, Darmstadt, Germany); HEK293 cells from the German Collection of Microorganisms and Cell Culture GmbH (DSMZ, Braunschweig, Germany). The Bac-to-Bac∗ baculovirus expression system was purchased from Invitrogen Life Technologies (ThermoFisher, Schwerte, Germany). Oligonucleotide synthesis was performed at BioTez Berlin Buch GmbH (Berlin, Germany). Nucleic acid sequencing was carried out at Eurofins MWG Operon (Ebersberg, Germany). HPLC grade methanol, acetonitrile, n-hexane, 2-propanol, and water were from Fisher Scientific (Waltham, MA, USA).

### 4.2. Database Searches and Amino Acid Alignments

For our initial database search, we screened the NCBI protein database with the key words “bony fish and lipoxygenase”. To exclude incomplete sequences, we eliminated all sequences involving less than 400 and more than 750 amino acids. Among the remaining sequences, we obtained 7 hits annotated as arachidonic acid 15-lipoxygenating enzymes (XP_015813570.1, XP_005753048.1, XP_018588735.1, NP_955912.1, XP_005945486.1, XP_003966824.1, XP_019726606.1). One of them (NP_955912.1) was the zebrafish ALOX1, which has previously been characterized as AA 12S-lipoxygenating enzyme [28]. To explore whether the remaining 6 sequences represent authentic ALOX-isoforms, we put them through our multistep filtering strategy for authentic ALOX sequences [11]. This strategy involved three consecutive filtering processes: (i) Presence of the ALOX characteristic W-XXX-AK motif [51]. (ii) Presence of the conserved amino acid sequences representing the two iron ligand clusters [11]. (iii) Distance of the two iron ligand clusters [11]. In one ALOX-isoform (XP_019726606.1), the two iron ligand clusters were incomplete, so we excluded it from further characterization. Since all 5 remaining bony fish ALOX-isoforms survived the filtering procedure, we performed dual amino acid alignments using the human ALOX15 as template (https://www.ebi.ac.uk/Tools/psa/emboss_needle/ accessed on 25 April 2019). Here, we found that in one bony fish ALOX sequence (XP_003966824.1), several amino acids of the catalytic domain were lacking and this sequence was also excluded from further characterization. In contrast, the putative ALOX15 sequences of *Nothobranchius furzeri* (XP_015813570.1), *Pundamilia nyererei* (XP_005753048.1), *Scleropages formosus* (XP_018588735.1), and *Haplochromis burtoni* (XP_005945486.1) were complete and thus, constituted candidates for expression and enzymatic characterization. However, the putative ALOX15 sequence of *Haplochromis burtoni* (XP_005945486.1) was later on removed from the database and thus, this enzyme was eliminated from our list. Thus, 3 putative bony fish ALOX15 orthologs were left for experimental characterization: (i) *Nothobranchius furzeri* (XP_015813570.1), (ii) *Pundamilia nyererei* (XP_005753048.1), (iii) *Scleropages formosus* (XP_018588735.1). It should be stressed at this point that the original putative ALOX15 sequence of *S. formosus* (XP_018588735.1) was updated in May 2019 in the database and this enzyme is now annotated as AA 5-lipoxygenating enzyme (XP_018588735.2).

### 4.3. Structural Modelling and Model Validation

The protein sequences for the bony fish ALOXs isoforms (XP_015813570.1, XP_005753048.1, XP_018588735.1, NP_955912.1) and for human ALOX12 (NP_000688.2) were downloaded from the NCBI protein database. The Prime Module of Schrodinger suite was used for generating the 3D models for these proteins. The suitable template structures for generating the 3D models were identified using BLAST search against a protein data bank (https://www.rcsb.org/, US data center for the open access global Protein Data Bank). Rabbit ALOX15 (PDB ID: 2P0M, Chain A) and human ALOX15B (PDB id: 4NRE) were selected as templates based on sequence similarity and two 3D structures were generated for each enzyme. The stereo chemical properties of the 3D models were evaluated by generating Ramachandran plots using PROCKECK and nonbonded atomic interaction patterns with ERRAT (Appendix A). The root-mean-square deviation (RMSD) between templates and generated models were calculated to quantify the degree of structural similarity.

### 4.4. Cloning and Expression of Bony fish ALOX-Isoforms

The coding sequences extracted from the database were chemically synthesized (BioCat GmbH, Heidelberg, Germany), further subcloned and expressed as described [8]. A more detailed description of the procedures is given in the methodological supplement (Appendix A).

### 4.5. Mutagenesis Studies

A detailed description of the procedures is given in the methodological supplement (Appendix A).

### 4.6. In Vitro Activity Assays and HPLC Analyses of the Reaction Products

A detailed description of the procedures is given in the methodological supplement (Appendix A). For control activity assays, a lysate supernatant of *E. coli* cells was used, which was transformed with a control expression plasmid. A similar experiment was carried out using the cellular supernatant of Sf9 infected with a non-lipoxygenase baculovirus.

### 4.7. LC-MS/MS Analysis of the Reaction Products

After incubation of the different bony fish ALOX15 isoforms with arachidonic acid, the oxygenation products were prepared by RP-HPLC and analyzed by LC-MS/MS.

### 4.8. Membrane Oxygenase Activity of Bony Fish ALOX-Isoforms

Different volumes of the cell-free supernatants were incubated with submitochondrial particles in PBS as described. Reaction products were reduced, total lipids were extracted [38], hydrolyzed under alkaline conditions, and analyzed by RP-HPLC. The hydroxy PUFA/PUFA ratio was determined comparing the peak areas of the major polyenoic fatty acids (LA + AA) and the conjugated dienes formed.

### 4.9. Experiments with Heavy Oxygen Isotops

To prove the incorporation of atmospheric oxygen during AA oxygenation by the putative bony fish ALOX15 orthologs, we carried out AA oxygenase activity assays under ^18^O_2_-atmosphere. Ten mL PBS containing 20 µM AA was extensively flushed with argon to remove the bulk of ^16^O_2_. Then, 500 µL of this anaerobized reaction mixture was transferred to a 0.5 mL Eppendorf tube, which was previously filled with argon gas. Next, the assay mixture in the Eppendorf tube was equilibrated with ^18^O_2_ gas (Sigma-Aldrich, St. Louis, MO, USA; 99% isotopic purity) and the ALOX reaction was initiated by the addition of enzyme (10–20 µL of cellular lysis supernatants). After a 3 min incubation period, the hydroperoxy fatty acids were reduced to the corresponding alcohols by the addition of solid sodium borohydride, reaction products (12*S*-HETE and 8*R*-HETE) were prepared by combined NP/CP-HPLC, and the ratio of the mass ions at *m*/*z* 319 (incorporation of one ^16^O atom into the reaction product) and *m*/*z* 321 (incorporation of one ^18^O atom into the reaction product) was quantified by LC-MS for both 12*S*- and 8*R*-HETE.

### 4.10. Miscellaneous Methods

For SDS-PAGE analysis, 100 µg denatured protein of the bacterial lysate supernatants were run on a 7.5% polyacrylamide gel. The proteins were then transferred onto a Protran BA 85 Membrane (Carl Roth GmbH, Karlsruhe, Germany) and the blots were probed with an anti-His-HRP antibody (Miltenyi Biotec GmbH, Bergisch Gladbach, Germany). Immunoreactive bands were visualized using the SERVALight Polaris CL HRP WB Substrate Kit (Serva Electrophoresis GmbH, Heidelberg, Germany). Chemiluminescence was detected on a FUJIFILM Luminescent Image Analyzer LAS-1000plus & Intelligent Dark Box II. The protein concentrations in the bacterial lysates were quantified using Bradford Reagent for quantitative protein determination (AppliChem, VWR International GmbH, Darmstadt, Germany) according to the instructions of the vendor. Amino acid sequence alignments were performed using the EMBOSS Needle tool (https://www.ebi.ac.uk/Tools/psa/emboss_needle/ accessed on 25 April 2019).

### 4.11. Statistical Evaluation of the Experimental Raw Data 

Statistical calculations and figure design were performed using GraphPad prism version 8.00 for Windows (GraphPad Software, La Jolla, CA, USA, www.graphpad.com).

## 5. Conclusions

Although the putative bony fish ALOX15 orthologs cannot be distinguished from corresponding mammalian enzymes on the basis of their structural characteristics, they have fundamentally different catalytic properties. The bony fish enzymes exhibit a narrow substrate specificity, lack membrane oxygenase activities, show a different kind of dual reaction specificity, and do not follow the Triad Concept. These functional differences suggest a targeted optimization of the catalytic properties of ALOX15 orthologs during vertebrate development.

## Figures and Tables

**Figure 1 ijms-24-14154-f001:**
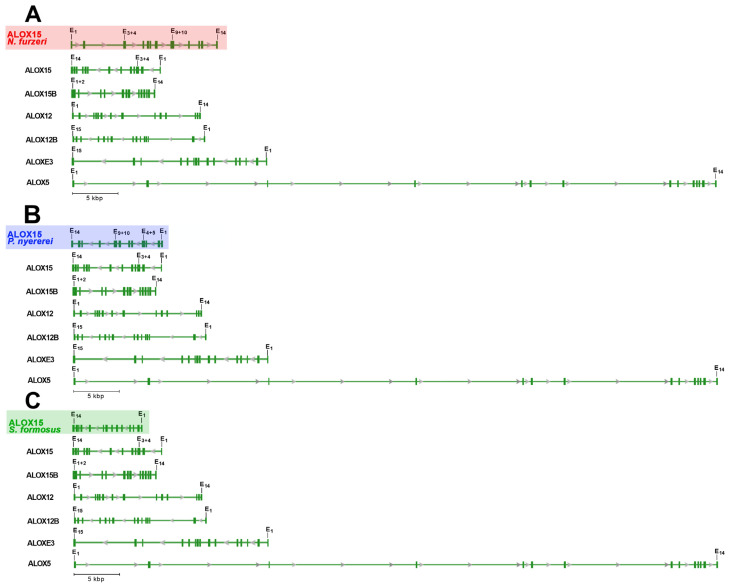
Exon-intron structure of the putative bony fish ALOX15 genes and structural comparison with the 6 different human ALOX-isoforms. (**A**) Structural comparison of the putative ALOX15 gene of *N. furzeri* with the genes encoding for human ALOX15, ALOX15B, ALOX12, ALOX12B, and ALOX5. (**B**) Structural comparison of the putative ALOX15 gene of *P. nyererei* with the genes encoding for human ALOX15, ALOX15B, ALOX12, ALOX12B and ALOX5. (**C**) Structural comparison of the putative ALOX15 gene of *S. formosus* with the genes encoding for human ALOX15, ALOX15B, ALOX12, ALOX12B, and ALOX5. The untranslated regions of E1 and E14 are not shown.

**Figure 2 ijms-24-14154-f002:**
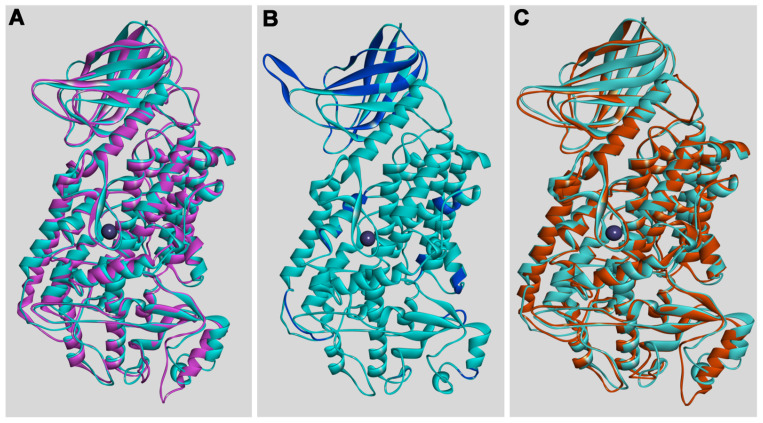
The 3D structure of the putative ALOX15 of *N. furzeri* exhibits a high degree of strutural similarity with (**A**) human ALOX15B, (**B**) rabbit ALOX15, and (**C**) human ALOX12. Three-dimensional structural models were worked out for the putative bony fish ALOX15 orthologs on the basis of the X-ray coordinates of human ALOX15B [32] as described in the Materials and Methods section. The model of the *N. furzeri* enzyme (turquoise) was overlaid with the 3D strutures of rabbit ALOX15 (mangenta, PDB: 2P0M), human ALOX15B (blue, PDB: 4NRE), and human ALOX12 (brown, home-made model).

**Figure 3 ijms-24-14154-f003:**
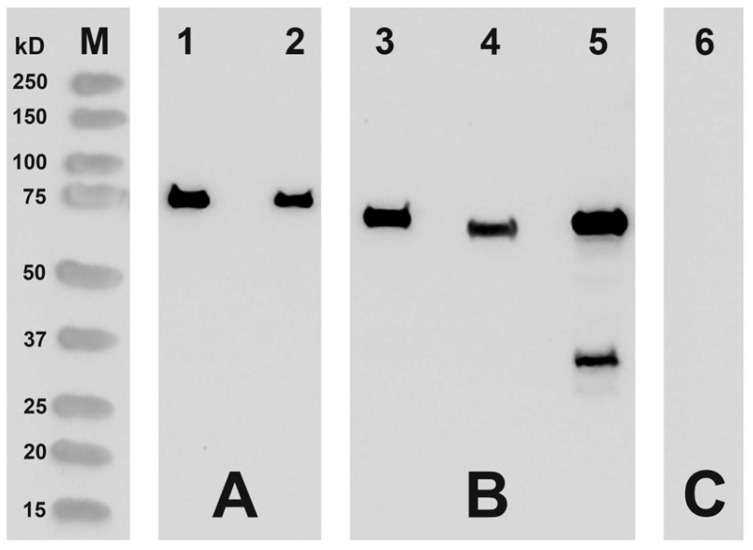
Expression of the putative bony fish ALOX15 orthologs in the baculovirus insect cell system. Putative ALOX15 orthologs of selected bony fish species were expressed as N-terminal His-tag fusion proteins in Sf9 cells and aliquots of the cell lysate supernatants were analyzed by Western-blot analysis using an anti-his-tag fusion protein antibody. (M) Molecular weight markers. (**A**) Quantification of the immune signals. Here, known amounts (lane 1, 500 ng; lane 2, 250 ng) of pure recombinant N-terminal his-tag fusion ALOX of *Myxococcus fulvus* were applied. (**B**) Expression of putative bony fish ALOX15 orthologs (lanes 3–5). Lane 3: putative ALOX15 ortholog of *N. furzeri*, 100 µg lysate supernatant protein; lane 4, putative ALOX15 ortholog of *P. nyererei*, 100 µg lysate supernatant protein; lane 5, putative ALOX15 ortholog of *S. formosus*, 100 µg lysate supernatant protein. Expressing the putative ALOX15 ortholog of *S. formosus*, we observed an additional immunoreactive band with a molecular weight of about 30 kDa (lane 5). The identity of this additional band remains unclear. (**C**) Sf9 cell lysate supernatant (100 µg total lysate supernatant protein) that was infected with the non-lipoxygenase control baculovirus (lane 6).

**Figure 4 ijms-24-14154-f004:**
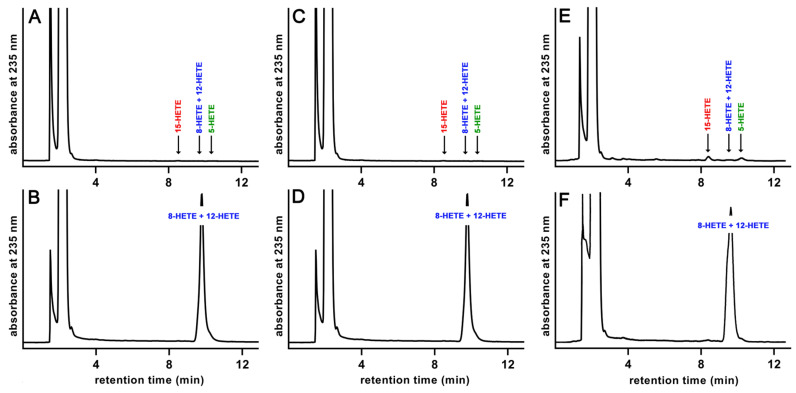
RP-HPLC analysis of the AA oxygenation products formed by the putative bony fish ALOX15 orthologs. The recombinant enzymes (aliquots of the cellular lysate supernatants) were incubated in PBS with 100 µM AA for 3 min and the AA oxygenation products were analyzed by RP-HPLC. (**A**) No-enzyme control (PBS), (**B**) *N. furzeri*, (**C**) heat control, *N. furzeri*, (**D**) *P. nyererei*, (**E**) Non-ALOX baculovirus infection, (**F**) *S. formosus*.

**Figure 5 ijms-24-14154-f005:**
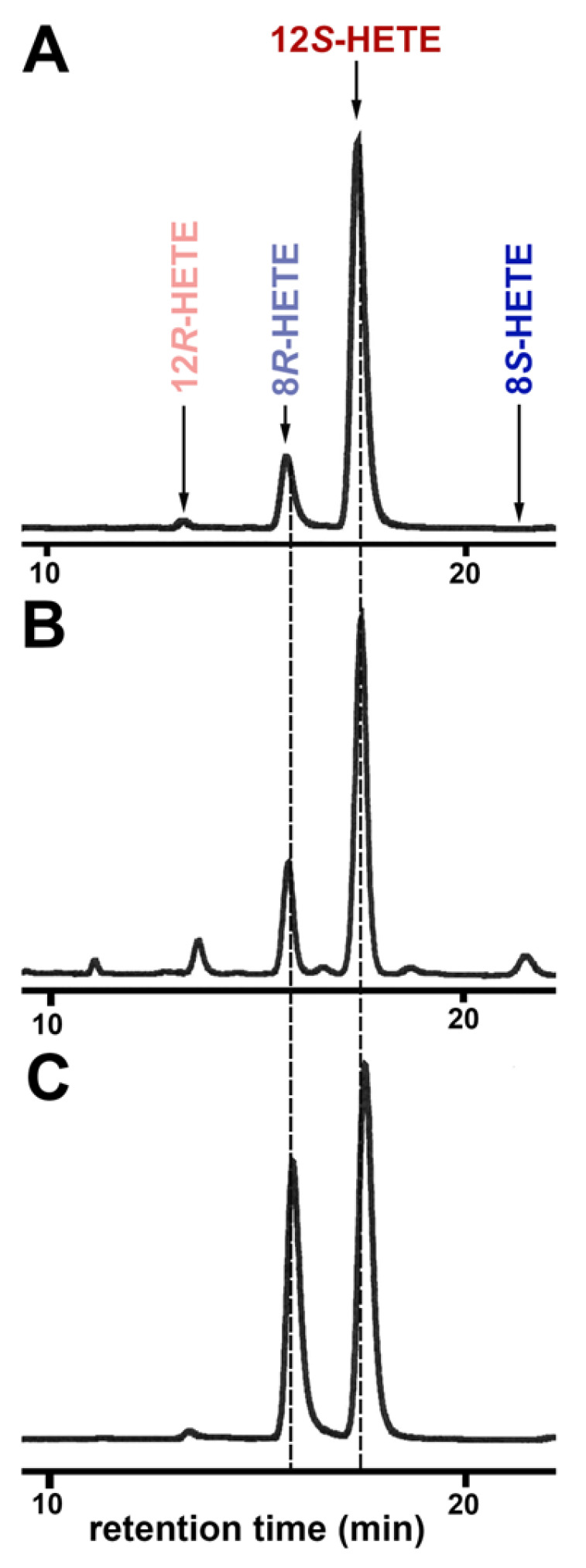
Partial NP/CP-HPLC chromatograms indicating the enantiomer composition of the ALOX products. AA oxygenation products formed by the putative bony fish ALOX15 orthologs were prepared by RP-HPLC and further analyzed by combined NP/CP-HPLC separating the positional and optical isomers of 12- and 8-HETE. (**A**) ALOX15 ortholog of *N. furzeri.* (**B**) ALOX15 ortholog of *P. nyererei.* (**C**) ALOX15 ortholog of *S. formosus*. The retention time of the authentic standards are indicated by the arrows above the traces.

**Figure 6 ijms-24-14154-f006:**
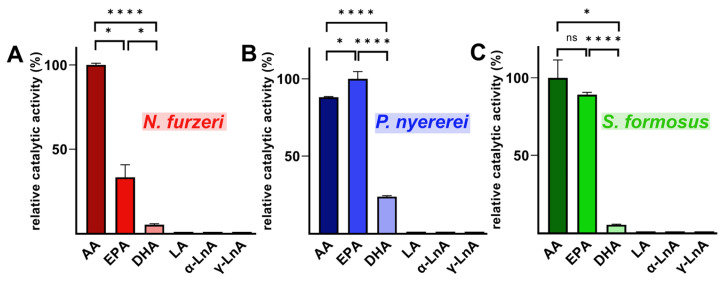
Substrate specificity of the putative bony fish ALOX15 orthologs. Activity assays were carried out with different fatty acids (50 µM). Aliquots of the cellular lysate supernatants were used as enzymes source. Two independent activity assays were carried out for each fatty acid substrate and each enzyme. Since the C_18_ fatty acids were hardly metabolized, their oxygenation rates were set 1% of the oxygenation rates of the most suitable substrate for each enzyme. Two identical samples were run for each fatty acid (biological replicates) and each sample was analyzed twice (technical replicates). (**A**) *N. furzeri*, (**B**) *P. nyererei,* (**C**) *S. formosus*. Statistics: Panel A: AA vs. EPA: unpaired U-test, AA vs. DHA: unpaired *t*-test, EPA vs. DHA: unpaired U-test. Panel B: unpaired *t*-test for all comparisons. Panel C: AA vs. EPA: unpaired U-test, AA vs. DHA: unpaired U-test, EPA vs. DHA: unpaired *t*-test. Abbreviations: AA, arachidonic acid; EPA, 5,8,11,14,17-eicosapentaenoic acid; DHA, 4,7,10,13,16,19-docosahexaenoic acid; LA, linoleic acid; α-LnA, alpha-linolenic acid; γ-LnA, gamma linolenic acid. ns = *p* > 0.05, * *p* = ≤ 0.05, **** *p* = ≤ 0.0001.

**Figure 7 ijms-24-14154-f007:**
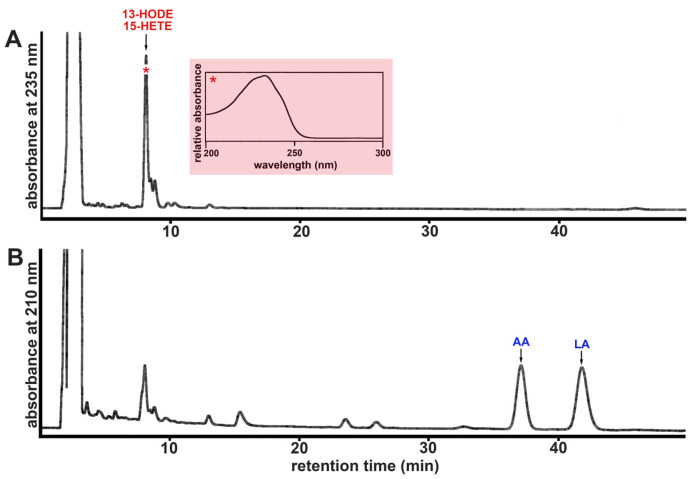
Membrane oxygenase activity of rabbit ALOX15. Purified rabbit ALOX15 was incubated in 0.5 mL PBS with submitochondrial particles (1.4 mg membrane protein/mL). Reaction products were reduced by the addition of solid borohydride, the sample was acidified, the total lipids were extracted [38], the lipid extracts were hydrolyzed under alkaline conditions, and the hydrolysates were analyzed by RP-HPLC (see Materials and Methods) recording the absorbances at 235 nm (OH-PUFAs, panel (**A**)) and at 210 nm (non-oxygenated PUFAs, panel (**B**)). The OH-PUFA/PUFA ratio was calculated as measure for the oxidation degree of the membrane lipids. Calibration curves (6-point measurements) for LA, AA, and 13-HODE were established to convert peak area units into nmoles of the analytes. * UV spectrum of the peak 13-HODE/15-HETE.

**Figure 8 ijms-24-14154-f008:**
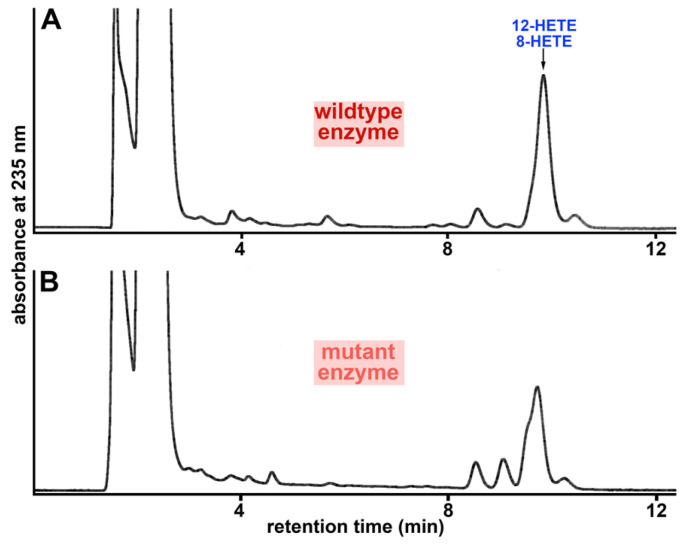
Reaction specificity of the putative ALOX15 ortholog of (**A**) *N. furzeri* and (**B**) its Val424Ile + Val425Met double mutant. Wildtype *N. furzeri* ALOX15 and its Val424Ile + Val425Met double mutant (cellular lysate supernatants) were incubated with AA in PBS for 3 min and the reaction products were analyzed by RP-HPLC.

**Figure 9 ijms-24-14154-f009:**
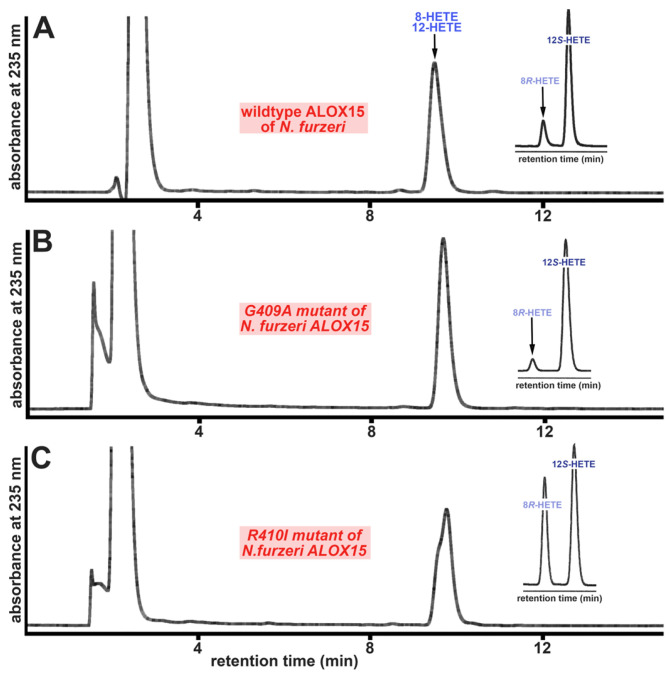
Reaction specificity of wildtype ALOX15 ortholog of *N. furzeri* and its Gly410Ala and Arg409Ile mutants. Enzyme preparations (cellular lysate supernatants of Sf9 cells) were incubated with AA in PBS for 5 min. The reaction products were reduced and analyzed by RP-HPLC as described in Materials and Methods. Insets: The conjugated dienes were prepared by RP-HPLC and further analyzed by combined NP/CP-HPLC (see Materials and Methods). (**A**) Wildtype enzyme, (**B**) Gly410Ala mutant, (**C**) Arg409Ile mutant.

**Table 1 ijms-24-14154-t001:** Putative ALOX15 orthologs in bony fish. A search of the NCBI protein database using the key words “lipoxygenase” and “bony fish” suggested the presence of these enzymes in seven of the 54 different bony fish species. Since the sequences for *T. rubripes* and *H. comes* were incomplete (Appendix A), they did not pass our filtering strategy.

Acession Number	Species	Amino Acids
NP_955912.1	*Danio rerio*	670
XP_015813570.1	*Nothobranchius furzeri*	670
XP_005753048.1	*Pundamilia nyererei*	670
XP_018588735.1	*Scleropages formosus*	668
XP_005945486.1	*Haplochromis burtoni*	682
XP_003966824.1	*Takifugu rubripes*	620
XP_019726606.1	*Hippocampus comes*	433

**Table 2 ijms-24-14154-t002:** Nucleotide sequence identity of the putative ALOX15 genes of the selected bony fish species with the different human ALOX genes. The nucleotide sequences of the putative ALOX15 cDNAs were extracted from the database and nucleotide sequence alignments were carried out using the EMBOSS Needle tool (https://www.ebi.ac.uk/Tools/psa/emboss_needle/ accessed on 25 April 2019). Bold letters and grey background mark average values.

Species	Nucleotide Sequence Identity with Human ALOX-Isoforms (%)					
	ALOX15	ALOX15B	ALOX12	ALOX12B	ALOXE3	ALOX5
*D. rerio*	32	31	38	37	43	20
*N. furzeri*	22	22	35	34	36	22
*P. nyererei*	37	36	40	39	35	16
*S. formosus*	39	39	38	39	33	15
**Average**	**33**	**32**	**38**	**37**	**37**	**18**

**Table 3 ijms-24-14154-t003:** Structural similarity scores obtained for pairwise comparison of the 3D models of the putative bony fish ALOX15 orthologs with the structures of rabbit ALOX15, human ALOX15B and human ALOX12. Homology models of the putative bony fish ALOX15 orthologs were worked out on the basis of the X-ray coordinates of human ALOX15B (4NRE, upper part of the Table) and rabbit ALOX15 (2P0M, lower part of the Table). The global numeric similiarity scores were calculated as described in the Materials and Methods section.

	Rabbit ALOX15	Human ALOX15B	Human ALOX12
3D models constructed on the crystal structure of human ALOX15B (4NRE)			
*N. furzeri*	3.176	1.022	1.326
*P. nyererei*	3.114	1.295	1.674
*S. formosus*	3.150	0.981	1.324
*D. rerio*	3.168	1.249	1.508
3D models constructed on the crystal structure of rabbit ALOX15 (2P0M)			
*N. furzeri*	1.658	3.301	1.133
*P. nyererei*	1.237	3.281	1.357
*S. formosus*	1.891	3.316	1.449
*D. rerio*	1.670	3.338	1.308

**Table 4 ijms-24-14154-t004:** Expression levels of the putative ALOX15 orthologs of bony fish species. The expression levels of the putative bony fish ALOX15 orthologs were calculated from the intensities of the immune signals using the purified *M. fulvus* ALOX as reference compound. This enzyme was previously expressed as N-terminal his-tag fusion protein and was subsequently purified to electrophoretic homogeneity by affinity chromatography on a Ni-NTA-agarose column.

Species	ALOX Protein (mg/L Expression Culture)	Molecular Weights of the ALOX Fusion Proteins (kDa)
*N. furzeri*	4.7	81.12
*P. nyererei*	1.2	81.25
*S. formosus*	2.8	80.85

**Table 5 ijms-24-14154-t005:** Incorporation of atmospheric oxygen into the AA oxygenation products formed by the putative bony fish ALOX15 orthologs. The methodological details for these experiments are provided in Section 4.9.

Species	^18^O/^16^O-Ratio (%) in the Reaction Product			
	12-HETE		8-HETE	
	^18^O	^16^O	^18^O	^16^O
*N. furzeri*	90.8	9.2	87.0	13.0
*P. nyererei*	89.9	10.1	90.3	9.7
*S. formosus*	91.8	8.2	89.0	11.0

**Table 6 ijms-24-14154-t006:** Biomembrane oxygenase activities of bony fish ALOX-isoforms. Aliquots of the cellular lysate supernatants (identical AA oxygenase activities) of the putative bony fish ALOX15 orthologs were incubated in PBS with submitochondrial membranes for 15 min. Sample workup and HPLC analysis were carried out as described in the legend to Figure 6. The OH-PUFA/PUFA ratios (in %) were calculated from the RP-HPLC chromatograms. Two identical samples were run for each enzyme preparation (biological replicates).

Lipoxygenase	OH-PUFA/PUFA Ratio (%)
No-enzyme control	0.05 ± 0.03
Rabbit ALOX15	5.60 ± 0.33
*N. furzeri*	0.04 ± 0.01
*P. nyererei*	0.21 ± 0.20
*S. formosus*	0

**Table 7 ijms-24-14154-t007:** Patterns of AA oxygenation products formed by putative bony fish ALOX15 ortholog mutants. Wildtype and mutant ALOX15 orthologs were expressed in *E. coli* (*S. formosus*) or Sf9 cells (*N. furzeri, P. nyererei*) and aliquots of the enzyme preparations (cellular lysate supernatants) were incubated with AA (100 µM) in PBS for 3 min. Conjugated dienes were prepared by RP-HPLC and further analyzed by combined NP/CP-HPLC.

	8*R*-HETE: 12*S*-HETE-Ratio (%)		
	Wildtype	Gly410Ala	Arg409Ile
*N. furzeri*	15:85	7:93	40:60
*P. nyererei*	21:79	0:100	41:59
*S. formosus*	43:57	0:100	51:49

## Data Availability

All data generated or analyzed during this study are included in this published article and in the Appendix A. Original experimental raw data can be obtained upon request from S.R., H.K. and D.H.

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
