# Peer review of "Bony Fish Arachidonic Acid 15-Lipoxygenases Exhibit Different Catalytic Properties than Their Mammalian Orthologs, Suggesting Functional Enzyme Evolution during Vertebrate Development"

_ijms, 2023, doi:10.3390/ijms241814154_

Round 1
Reviewer 1 Report
This manuscript extends another recent paper, published in December 2022 in IJMS (ref.#31) (lines 550-557 of current manuscript). In the previous paper, three ALOX15 orthologs from three different bonny fishes were selected for study, cloned, expressed as recombinant proteins and characterized concerning their catalytic properties (temperature dependence, activation energy, pH-dependence, substrate affinity and substrate specificity with
different polyenoic fatty acids). It was then concluded that functional ALOX isoforms occur in bonny fish but their catalytic properties are different from those of mammalian enzymes.
In the current manuscript, the authors continue the characterization of the same three bonny fish ALOX15 orthologs. It is highlighted that, compared to the human ortholog, bonny fish enzymes exhibit a much more restrictive substrate specificity, lack membrane oxygenase activity, and display a different reaction specificity with arachidonic acid. In addition, mutagenesis studies indicate that the molecular principles explaining the reaction specificity of mammalian ALOX15 orthologs (the so-called Triad Concept) are not valid for bonny fish enzymes. The authors conclude that the functionality of these enzymes has been optimized during development (possibly they mean evolution).
MAJOR POINTS
A) To a certain extent, the current manuscript seems to ignore the 2022 IJMS article (despite lines 550-557). The authors dedicate considerable part of the current manuscript (Results section 2.1, including Table 1; and methods section 4.2) to explain how the three bonny fish ALOX15 orthologs were selected for study, and to the description of cloning and expression procedures (Sections 2.2 and 4.4). In fact, all of this seems to belong to the previous, 2022 article, since the selection and cloning of these orthologs was then performed. If this is so, parts of the current manuscript should be deleted and substituted just by reference to the previous article. If there are good reasons to re-describe these results and procedures, they should be made explicit.
B) The concept of “targeted optimization” used in the abstract line 28 and Conclusions line 766 should be explicitly explained in the Discussion.
MINOR POINTS
Line 18. The three studied species should be explicit. Full binomial names should be cited in italics.
The authors should follow the rule of using full binomial names at the first mention of a species in abstract, tables/figures and main text. Also, all the taxonomical names should italicized. This is not so in the manuscript, e.g. Table 2 and many other instances.
Line 52. What is meant by “principle occurrence (of) ALOX isoforms”?
Abbreviations should be defined at their first use. For instance see AA in line 68, or PUFA in line 326.
Line 39: Archaea, not Archeae
Lines 83-84: the “Triad Concept” needs a reference here.
Line 97: “search” not “serach”
Line 148: “created” not “cerated”
Lines 188-190: wrong spacing
Line 246: “conjugated” not “conjuagted”
Table 5: the long title of this Table belongs to Methods
Line 438: “confirmed” not “cofirmed”
Line 531: “one” not “on”
Line 534: “development” should be probably replaced by “evolution”
Line 667: “coordinates” not “co-ordiantes”
The English is generally fine. Many typos.
Author Response
Comment of reviewer: A) To a certain extent, the current manuscript seems to ignore the 2022 IJMS article (despite lines 550-557). The authors dedicate considerable part of the current manuscript (Results section 2.1, including Table 1; and methods section 4.2) to explain how the three bonny fish ALOX15 orthologs were selected for study, and to the description of cloning and expression procedures (Sections 2.2 and 4.4). In fact, all of this seems to belong to the previous, 2022 article, since the selection and cloning of these orthologs was then performed. If this is so, parts of the current manuscript should be deleted and substituted just by reference to the previous article. If there are good reasons to re-describe these results and procedures, they should be made explicit.
Response of authors: It was by no means our intention to ignore the 2022 paper (Roigas S et al. Int J Mol Sci. 2022;23(24). Epub 2022/12/24. doi: 10.3390/ijms232416026.) and we have even referenced this paper in the original version of the present ms (ref. 31). Although the two papers characterize functional aspects of the same enzymes there is no thematic overlap between the two reports. Most importantly, the take home messages of the two papers is completely different since in the present paper we focus on functional aspects of ALOX15 evolution. This topic was not at all part of the 2022 paper. One of the major questiones we wanted to answer in the present paper was of whether or not ALOX15 orthologs do frequently occur in bony fish. For this purpose, we developed a data base searching strategy for ALOX isoforms involving three levels of stringecy. Applying this three-step searching strategy we found that among the completely sequenced 54 bony fish genomes only 7 (less than 15 %) involved annotated ALOX15 genes (Table 1). From these data we concluded that ALOX15-orthologs do less frequently occur in bony fish than in mammals. To allow other researchers to perform similar database searches we described our searching strategy in great detail in the present paper and we would like to keep this paragraph as it is. We also think that the structural data provide in Figures 1 + 2 and in Tables 2 + 3 provide novel (have not been published before) und useful information and thus, they should be kept in the ms. However, to respond to the critical remarks of this reviewer we referenced the 2022 paper earlier on in the ms (page 1, line 38) and eliminated methodological details of enzyme preparation from the paper. These details that partly overlap with the 2022 paper (Roigas S et al. Int J Mol Sci. 2022;23(24) are transfered to a methodological supplement.
Comment of reviewer: B) The concept of “targeted optimization” used in the abstract line 28 and Conclusions line 766 should be explicitly explained in the Discussion.
Response of authors: In response to this point of criticism we included a novel paragraph in the section 3 (Discussion) explaining why we think that the catalytic properties of ALOX15 orthologs were optimized during vertebrate evolution. This paragraph (page 15, lines 546-572), which is visualized in the graphical abstract, reads as follows: “When we explored the reaction specificity of a large number of mammalian ALOX15 orthologs we previously reported that this enzyme property has systematically been altered during mammalian evolution (Heydeck et al., The Reaction Specificity of Mammalian ALOX15 Orthologs is Changed During Late Primate Evolution and These Alterations Might Offer Evolutionary Advantages for Hominidae. Front Cell Dev Biol. 2022 Apr 21;10:871585. doi: 10.3389/fcell.2022.871585. PMID: 35531094; PMCID: PMC9068934.) In that paper we showed that mammals ranked in evolution lower than gibbons including the frequently used lab animals (mice and rats) express arachidonic acid 12-lipoxygenating Alox15 orthologs. In contrast, hominids (different human subspecies, chimpanzee, bonobos, gorillas, orangutans) express arachidonic acid 15-lipoxygenating enzymes. Although some mammalian species (rabbits, kangaroo rats) violate this concept more than 95 % of mammalian ALOX15 orthologs do follow it. In other words, most mammalian ALOX15 orthologs are arachidonic acid 12-lipoxygenating enzymes. The putative bony fish ALOX15 orthologs decribed here are also arachidonic acid 12-lipoxygenating enzymes and thus, in this respect they are more closely related to the enzymes of non-hominid mammals than to the hominid orthologs. Moreover, the bony fish enzymes exhibit a narrow substrate specificity. They strongly prefer C20 fatty acids but do hardly oxygenate C22 and C18 fatty acid derivatives (Figure 6). Thus, the substrate specificity of the ALOX15 orthologs was broadened during vertebrate evolution. Since C18 fatty acids (linoleic acid, linolenic acid) frequently occur in membranes of eucaryotic cells, bony fish ALOX15 orthologs were likely to exhibit a limited membrane oxygenase activity. On the other hand, mammalian ALOX15 orthologs are capable of oxygenating biomembranes [Kuhn et al., J Biol Chem. 1990;265(30):18351-61. PubMed PMID: 2120232; Takahashi et al., Eur J Biochem. 1993;218(1):165-71. PubMed PMID: 8243462]. Testing the membrane oxygenase activity of the three putative bony fish ALOX15 orthologs we found that these enzymes lack any membrane oxygenase activity. Thus, ALOX15 orthologs have apparently acquired a membrane oxygenase activity during vertebrate evolution. Although the driving forces for the evolutionary change in reaction specificity remains to be explored these functional alterations (broadening of substrate specificity, aquiring a membrane oxygenase activity) might have optimized the biological role of mammalian ALOX15 orthologs during the maturational breakdown of mitochondria during late erythropoiesis (Schewe et al., FEBS Lett. 1975;60(1):149-52. PubMed PMID: 6318).”
Comment of reviewer: Line 18. The three studied species should be explicit. Full binomial names should be cited in italics. The authors should follow the rule of using full binomial names at the first mention of a species in abstract, tables/figures and main text. Also, all the taxonomical names should italicized. This is not so in the manuscript, e.g. Table 2 and many other instances.
Response of authors: We thank the reviewer for this comment and corrected the ms accordingly.
Comment of reviewer: Line 52. What is meant by “principle occurrence (of) ALOX isoforms”?
Response of authors: We apologize for this typo (not principle but principal). This sentence basically means that ALOX-isoforms have previously been reported to be present in selected bony fish species but that it remained unclear how frequently ALOX15 orthologs occur in these animals. We modified the text accordingly (page 2, line 53-54 and lines 61-62).
Comment of reviewer: Abbreviations should be defined at their first use. For instance see AA in line 68, or PUFA in line 326.
Response of authors: We double checked that all abbreviations were spelt out when they were used for the first time in the text. Moreover we inserted an abbreviation list, in which all non-standard abbreviations were summarized and spelt out (page 19, lines 778-786).
Comment of reviewer: Typos line 39: Archaea, not Archeae, line 97: “search” not “serach”, line 148: “created” not “cerated”, lines 188-190: wrong spacing, line 246: “conjugated” not “conjuagted”, line 438: “confirmed” not “cofirmed”, line 531: “one” not “on”, line 534: “development” should be probably replaced by “evolution”, line 667: “coordinates” not “co-ordiantes”
Response of authors: We apologize for these typos and corrected them as suggested by the reviewer.
Comment of reviewer: Lines 83-84: the “Triad Concept” needs a reference here.
Response of authors: We inserted two appropriate references (line 87-88).
Comment of reviewer: Table 5: the long title of this Table belongs to Methods
Response of authors: We removed the methodological details from the figure legends (page 8, lines 279-281).
Reviewer 2 Report
The manuscript submitted by Roigas .etal describe the enzyme evolution during vertebrate development using arachidonic acid 15-lipoxygenases from bony fish as research model, the results are rich, but some data are from modle prediction, and some experimental data such as enzyme activities lack specific quantitative analysis, hence the conclusion is not clear and should be rewrited. Some major issues are as listed as follows.
(1) The manuscript conducted some characterization work on the properties of ALOX15 orthologs from bony fish, while the article has not formed a definite conclusion, such as described in the manuscript “aken together, our data suggest a functional optimization of ALOX15 orthologs during vertebrate evolution”. This is not a definite conclusion, but a description of the results.
(2) The author has not determined the relevant structure, but simply made some predictions, and it is very unscientific to describe it as “we here determined additional structural and functional characteristics”.
(3) Figure 2. The 3D- structure of the putative ALOX15, The key residues or regions are not marked in the figure, and the prediction description in the method section is not described in detail, in addition, the reliability of the model is not verified.
(4) Line216, “immune affinity chromatography on a Ni-agarose column” is not appropriate, is it “metal affinity chromatography”?
(5) Some specific data of the relative activity of the enzymes should be provided as umol/mg/min, so as to facilitate the comparison between different studies.
Author Response
Comment of reviewer: The manuscript conducted some characterization work on the properties of ALOX15 orthologs from bony fish, while the article has not formed a definite conclusion, such as described in the manuscript “taken together, our data suggest a functional optimization of ALOX15 orthologs during vertebrate evolution”. This is not a definite conclusion, but a description of the results.
Response of authors: With all due respect for the expertise of the reviewer we disagree with this point of criticism. In this paper we characterize structural and functional properties of putative ALOX15 orthologs in three different bony fish species and the experimental results are clearly described in the Results section (chapters 2.1- 2.8). We found that despite the high degree of structural similarity between bony fish and mammalian ALOX15 orthologs (Figures 1 + 2, Tables 1, 2, 3), the functional characteristics of the two enzyme classes were surprisingly different. These differences prompted us to conclude that the functionality of ALOX15 orthologs was optimized during vertebrate evolution. To respond to the criticism of the reviewer we inserted an additional paragraph into section 3 (Discussion), in which we describe this situation in more detail. This paragraph (page 15, lines 546-572), which is also visualized in the graphical abstract, reads as follows: “When we explored the reaction specificity of a large number of mammalian ALOX15 orthologs we previously reported that this enzyme property has systematically been altered during mammalian evolution (Heydeck et al., The Reaction Specificity of Mammalian ALOX15 Orthologs is Changed During Late Primate Evolution and These Alterations Might Offer Evolutionary Advantages for Hominidae. Front Cell Dev Biol. 2022 Apr 21;10:871585. doi: 10.3389/fcell.2022.871585. PMID: 35531094; PMCID: PMC9068934.) In that paper we showed that mammals ranked in evolution lower than gibbons including the frequently used lab animals (mice and rats) express arachidonic acid 12-lipoxygenating Alox15 orthologs. In contrast, hominids (different human subspecies, chimpanzee, bonobos, gorillas, orangutans) express arachidonic acid 15-lipoxygenating enzymes. Although some mammalian species (rabbits, kangaroo rats) violate this concept more than 95 % of mammalian ALOX15 orthologs do follow it. In other words, most mammalian ALOX15 orthologs are arachidonic acid 12-lipoxygenating enzymes. The putative bony fish ALOX15 orthologs decribed here are also arachidonic acid 12-lipoxygenating enzymes and thus, in this respect they are more closely related to the enzymes of non-hominid mammals than to the hominid orthologs. Moreover, the bony fish enzymes exhibit a narrow substrate specificity. They strongly prefer C20 fatty acids but do hardly oxygenate C22 and C18 fatty acid derivatives (Figure 6). Thus, the substrate specificity of the ALOX15 orthologs was broadened during vertebrate evolution. Since C18 fatty acids (linoleic acid, linolenic acid) frequently occur in membranes of eucaryotic cells, bony fish ALOX15 orthologs were likely to exhibit a limited membrane oxygenase activity. On the other hand, mammalian ALOX15 orthologs are capable of oxygenating biomembranes [Kuhn et al., J Biol Chem. 1990;265(30):18351-61. PubMed PMID: 2120232; Takahashi et al., Eur J Biochem. 1993;218(1):165-71. PubMed PMID: 8243462]. Testing the membrane oxygenase activity of the three putative bony fish ALOX15 orthologs we found that these enzymes lack any membrane oxygenase activity. Thus, ALOX15 orthologs have apparently acquired a membrane oxygenase activity during vertebrate evolution. Although the driving forces for the evolutionary change in reaction specificity remains to be explored these functional alterations (broadening of substrate specificity, aquiring a membrane oxygenase activity) might have optimized the biological role of mammalian ALOX15 orthologs during the maturational breakdown of mitochondria during late erythropoiesis (Schewe et al., FEBS Lett. 1975;60(1):149-52. PubMed PMID: 6318).”
Comment of reviewer: The author has not determined the relevant structure, but simply made some predictions, and it is very unscientific to describe it as “we here determined additional structural and functional characteristics”.
Response of authors: This statement is at least in part not correct. For this ms we determined experimentally a number of functional characteristics of the three bony fish ALOX15 orthologs: i) We did determine experimentally (Figure 4+5) that these enzymes convert arachidonic acid to a mixture of 12S- and 8R-HETE. Thus, the enzymes exhibit a different kind of dual reaction specificity than mammalian ALOX15 orthologs. ii) We did determine experimentally (Table 5) that the three enzymes incorporate atmospheric oxygen but not water oxygen into their reaction products. iii) We did determine experimentally (Figure 6) that the three bony fish enzymes exhibit a narrow substrate specificity, which contrasts the broad substrate specificity of mammalian ALOX15 orthologs. iv) We did determine experimentally (Table 6) that in contrast to mammalian ALOX15 orthologs the three bony fish enzymes lack any membrane oxygenase activity. v) We did determine experimentally (Figure 8) that the bony fish enzymes do not follow the Triad Concept that explains the reaction specificity of all mammalian ALOX15 orthologs tested so far. vi) We did determine experimentally (Figure 9, Table 8) that the three bony fish enzymes only partly follow the A-vs-G Concept which explains the enantioselectivity of mammalian ALOX15 orthologs.
However, the reviewer is correct that we did not determine the structural properties of the bony fish ALOX15 orthologs. The gene and cDNA sequences were previously determined and we retrieved this data from public data bases. However, for the first time we compared these structural data with those of mammalian ALOX15 orthologs. This comparison prompted the conclusion that the three putative bony fish ALOX15 orthologs share a high degree of structural similarity with the mammalian ALOX15 orthologs. Moreover, on the basis of these sequence data we generated 3D-models for the corresponding proteins using standard algorhythms and compared the resulting 3D-models of the bony fish enzymes with those of the experimentally determined 3D structures of mammalian ALOX-isoforms. According to our opinion this is a very useful, reliable and scientific approach.
To respond to this comment of the reviewer we modified the text of the corresponding sentence. It now reads (page 2, lines 80-89): “To explore whether these enzymes constitute functional equivalents of mammalian ALOX15 orthologs, we here determined experimentally additional functional properties (enantioselectivity, substrate specificity, membrane oxygenase activity) of the three bony fish enzymes and compared selected structural enzyme characteristics of these proteins with those of the corresponding mammalian proteins. Despite the high degree of structural similarity between the putative bony fish ALOX15 orthologs and the mammalian enzymes we observed pronounced functional differences.”
Comment of reviewer: Figure 2. The 3D- structure of the putative ALOX15, the key residues or regions are not marked in the figure, and the prediction description in the method section is not described in detail, in addition, the reliability of the model is not verified.
Response of authors: 3D structural models of the three putative bony fish ALOX15 orthologs were constructed on the basis of the amino acid sequence data we retrieved from the databases. When high resolution X-ray data exist for corresponding enzymes from other species (this is the case for a large number of plant and animal ALOX-isoforms) this approach is a reliable standard procedure and the corresponding software tools have been used in a large number of previous studies. Because of the high degree of overall structural similarity (almost perfect superposition of the different 3D models in Figure 2 it makes no sense and mignht be even confusing when we label key residues such as the iron ligands or the catalytic center. Thus, we would like to keep Figure 2 as it is. However, we follow the advice of the reviewer and briefly discuss the reliability of the modeling procedure in section 4 (page 18, 705-708) and added two Tables to the supplement (Table S5 + S6) demonstrating this.
Comment of reviewer: Line216, “immune affinity chromatography on a Ni-agarose column” is not appropriate, is it “metal affinity chromatography”?
Response of authors: This criticism is justified since Ni-agarose chromatogphy is not an immune affinity chromatography. We think that “affinity chromatography on Ni-NTA agarose” is the right description for our method and we modified the ms accordingly (page 6, lines 220-221).
Comment of reviewer: Some specific data of the relative activity of the enzymes should be provided as umol/mg/min, so as to facilitate the comparison between different studies.
Response of authors: For our activity assays we used crude cell lysate supernatants as enzyme source. The protein content of the lysate supernants strongly depends on the concentrations of non-ALOX proteins and these concentrations strongly vary from preparation to preparation. Thus, we doubt that giving the catalytic activities in µmol product formation per mg total protein and min provides relevant information. In fact, quantifying the catalytic activity of recombinant ALOX preparations are only meaningful and can be compared when pure enzyme preparations are used and the catalytic activity is normalized for the iron content of the enzyme preparations.
Round 2
Reviewer 1 Report
The changes requested by this reviewer have been satisfactorily attended.
The english is fine, but attention should be given to possible typos and other minor edits
Reviewer 2 Report
The author has carefully revised the manuscript, resulting in a significant improvement in its quality. The revised article provides a clearer explanation of the research background and purpose in the introduction section, and accurate conclusion section. The manuscript can be accepted in present form.